# A Non-Systemic Phosphodiesterase-5 Inhibitor Suppresses Colon Proliferation in Mice

**DOI:** 10.3390/ijms24119397

**Published:** 2023-05-28

**Authors:** Avelina Lee, Iryna Lebedyeva, Wenbo Zhi, Vani Senthil, Herjot Cheema, Michael W. Brands, Weston Bush, Nevin A. Lambert, Madeline Snipes, Darren D. Browning

**Affiliations:** 1Department of Biochemistry and Molecular Biology, Augusta University, Augusta, GA 30912, USA; 2Department of Chemistry and Physics, Augusta University, Augusta, GA 30912, USA; 3Center for Biotechnology and Genomic Medicine, Augusta University, Augusta, GA 30912, USA; 4Department of Physiology, Augusta University, Augusta, GA 30912, USA; 5Department of Pharmacology and Toxicology, Augusta University, Augusta, GA 30912, USA

**Keywords:** phosphodiesterase inhibitor, pharmacokinetics, intestines, colon neoplasms, cyclic guanosine monophosphate

## Abstract

Phosphodiesterase-5 inhibitors (PDE5i) are under investigation for repurposing for colon cancer prevention. A drawback to conventional PDE5i are their side-effects and drug–drug interactions. We designed an analog of the prototypical PDE5i sildenafil by replacing the methyl group on the piperazine ring with malonic acid to reduce lipophilicity, and measured its entry into the circulation and effects on colon epithelium. This modification did not affect pharmacology as malonyl-sildenafil had a similar IC_50_ to sildenafil but exhibited an almost 20-fold reduced EC_50_ for increasing cellular cGMP. Using an LC-MS/MS approach, malonyl-sildenafil was negligible in mouse plasma after oral administration but was detected at high levels in the feces. No bioactive metabolites of malonyl-sildenafil were detected in the circulation by measuring interactions with isosorbide mononitrate. The treatment of mice with malonyl-sildenafil in the drinking water resulted in a suppression of proliferation in the colon epithelium that is consistent with results previously published for mice treated with PDE5i. A carboxylic-acid-containing analog of sildenafil prohibits the systemic delivery of the compound but maintains sufficient penetration into the colon epithelium to suppress proliferation. This highlights a novel approach to generating a first-in-class drug for colon cancer chemoprevention.

## 1. Introduction

Cancers of the colorectum are of the most significance due to the fact that related diagnoses and deaths reach almost 10% of all cancers. In the United States, the probability of a CRC diagnosis is approximately 1 in 23, but there are millions with an up to 5-fold higher risk, including those with recurrent polypectomy and those with first-person relatives with an early diagnosis [1]. These high-risk patients are often instructed to implement lifestyle changes and undergo surveillance colonoscopy to reduce their risk [2,3]. Colonoscopy is an effective approach to CRC prevention, but a number of socioeconomic reasons make it unavailable to many high-risk populations [4,5,6].

There is therefore an important clinical need for chemopreventive drugs, but there is currently nothing available except NSAIDs, which have a very limited efficacy and significant side effects [7]. Numerous preclinical studies support the idea that increasing cGMP levels in the colon epithelium can prevent CRC [8]. The source of epithelial cGMP in the intestine is a receptor guanylate cyclase (GC-C) that is activated by the endogenous secretagogues, guanylin and uroguanylin [9]. In vivo studies using knockout mice with defective cGMP signaling have shown that cGMP controls intestinal homeostasis by slowing epithelial turnover [10,11,12]. A loss of epithelial cGMP in Gucy2c^−/−^ mice results in increased intestinal carcinogenesis and treating mice with GC-C agonists can suppress disease in the Apc^Min/+^ model [13,14,15]. Another approach to increase cGMP in the colon epithelium is to block degradation using inhibitors of phosphodiesterase 5 (PDE5i) [16]. The treatment of mice with PDE5is, such as vardenafil and sildenafil, suppresses proliferation in the colon epithelium [15,17,18]. Landmark studies showed that a clinically relevant dose of sildenafil can block tumorigenesis by 50% in both AOM/DSS and Apc^Min/+^ mouse models of CRC [15,19]. Sildenafil reduced polyp multiplicity in these studies but did not affect the size of the initiated polyps, indicating that the central effect of cGMP is to suppress tumor initiation [8,20]. 

Two recent large cohort retrospective studies suggested that PDE5i might also prevent CRC in humans. The first report used the Swedish Hospital Discharge Register to show that polypectomy patients who had taken PDE5i had 36% reduced incidence of CRC compared to those who had never been exposed to PDE5i [21]. The second report interrogated the Veterans Affairs Database, and found that patients taking PDE5i over the long term had an up to 50% reduction in CRC incidence [22]. Prospective clinical studies are needed to support the repurposing of PDE5i for CRC chemoprevention. Conventional PDE5i (e.g., sildenafil, tadalafil) that are most widely prescribed for erectile dysfunction, have also been safely used to treat chronic diseases including pulmonary arterial hypertension and benign prostate hyperplasia [23,24]. However, side effects of these drugs such as headache, flushing, and dyspepsia, would be less tolerated by otherwise healthy patients using them to reduce their risk of CRC. Moreover, due to common predisposing factors, many high-risk CRC patients also take nitroglycerin to treat ischemic heart disease [25]. Due to drug–drug interactions, these patients could not benefit from PDE5i for CRC prevention. PDE5is that avoid systemic delivery would therefore be a major advantage as an agent for CRC chemoprevention. 

With the goal of developing a novel first-in-class agent for the primary chemopre-vention of CRC, the present study has designed a more hydrophilic analog of sildenafil containing a carboxylic acid residue. Our results suggest that this compound exhibits poor cell permeability, and that it avoids systemic delivery while retaining the anti-proliferative ability of PDE5i in the colon epithelium.

## 2. Results

### 2.1. A Carboxylic Acid Containing an Analog of Sildenafil Retains Pharmacological Activity but Exhibits Reduced Cell Entry

The lipophilic design of conventional PDE5is, such as sildenafil, tadalafil, and vardenafil, enables the entry into the circulation that is essential to reach the target tissues. To create a PDE5i with reduced systemic delivery, we sought to reduce lipophilicity by adding groups that carry a charge at physiological pH. To accomplish this, we substituted malonate for the methyl group on the piperazine ring of the prototypical PDE5i sildenafil. This structure (malonyl-sildenafil) is predicted to exhibit much higher solubility and reduced membrane permeability at physiological pH values (Figure 1A,B). An in vitro assay for PDE5i activity revealed an IC_50_ of 6.5 nM for malonyl-sildenafil that is similar to the parent compound (7.5 nM), indicating that the modification did not deleteriously affect the pharmacology (Figure 1C). We used a novel cell-based assay to assess the ability of the PDE5i to enter colon epithelial cells and activate cGMP signaling (Figure 2A). These results showed that malonyl-sildenafil was 18-fold less potent in terms of increasing cGMP levels in the cells than the parent compound based upon EC_50_ (Figure 2B). These observations were supported by a highly sensitive assay for the activation of cGMP-dependent protein kinase (Figure 2C).

### 2.2. Malonyl-Sildenafil Does Not Enter the Circulation Following Oral Administration in Mice

To determine the extent to which malonyl-sildenafil can enter the circulation, we developed an LC-MS/MS procedure to measure the compound in plasma (Figure 3). The ion spectrum used for sildenafil is based upon a previous study [26], and several common peaks were generated by malonyl-sildenafil including *m*/*z* at 283, 311 and 377. However, additional peaks were observed that are consistent with the loss of the carboxylic group (*m*/*z* = 503) and the loss of the whole malonyl group to generate desmethyl-sildenafil (*m*/*z* = 461). Tests one hour after oral administration of malonyl-sildenafil to mice at concentrations up to 36 mg/kg did not result in detectable levels in the plasma (Figure 4A). This contrasted with the sildenafil that was detected in plasma at 250 nM and 3.5 µM after administering 9 and 36 mg/kg, respectively. Notably, sildenafil was not detected in the feces collected between 2–4 h after oral administration, but malonyl-sildenafil was detected at high levels ranging from 90 µg/g to 244 µg/g after administering 9 and 36 mg/kg, respectively (Figure 4B). The decision to measure malonyl-sildenafil one hour following oral administration was based upon established pharmacokinetics for sildenafil [27,28]. The possibility that the reduced membrane permeability of malonyl-sildenafil could cause delayed entry into the circulation was tested by performing a time course using 9 mg/kg (Figure 4C). Apart from a very small (11 nM) peak at 1 h, these experiments revealed that there was no delayed peak, and malonyl-sildenafil was not detected in the plasma at any other time during the 24 h period examined. Notably, malonyl-sildenafil continued to be excreted at high levels in the feces 8 h after oral administration (Figure 4D).

Consuming PDE5i together with soluble guanylate cyclase stimulators such as riociguat, or nitrate activators such as isosorbide mononitrate (ISMN) and nitroglycerin, is contraindicated due to the risk of severe hypotension. Malonyl-sildenafil was not detected at high levels in the plasma compared to sildenafil, but bioactive metabolites could have been created that could enter the circulation undetected. To investigate this possibility, mice were treated with ISMN (300 mg/kg i.p.) alone, or together with orally administered sildenafil or malonyl-sildenafil (10 mg/kg). Arterial blood pressure was then measured over time using remote telemetry (Figure 5). ISMN, alone or with malonyl-sildenafil, slightly increased blood pressure from a baseline of around 100 mmHg to 126 mmHg and 117 mmHg, respectively. This was most likely due to the stress caused by the procedure. As expected, the co-administration of ISMN and sildenafil caused a rapid drop in blood pressure to 68 mmHg that returned to close to the baseline over the following 30 min.

### 2.3. Oral Administration of Malonyl-Sildenafil Suppresses Proliferation in the Colon Epithelium

The pharmacokinetic data support the idea that malonyl-sildenafil has reduced permeability and does not enter the circulation following oral administration in mice. It was therefore important to determine whether malonyl-sildenafil could enter the colon epithelium to suppress proliferation, which is thought to be the central effect of PDE5i that mediates colon cancer prevention [8,20]. Mice were maintained on malonyl-sildenafil (9 mg/kg) in the drinking water for up to 8 days, and there was no difference in body weight compared to those on sildenafil or water alone (Figure 6A). The initial drop in weight in the first couple of days most likely reflected a distaste for the compound since they initially drank less than the control groups. Mice were active and exhibited no observed deleterious effects during the time studied. Histological analysis of the colon mucosa after 8 days of consuming malonyl-sildenafil revealed no obvious epithelial damage, edema or leukocyte infiltration (Figure 6B). Malonyl-sildenafil was not detected in the plasma of mice drinking malonyl-sildenafil for 8 days, but low levels of sildenafil (3.3 nM) were observed in the plasma of mice drinking sildenafil (Figure 6C). To determine the effect of orally administered malonyl-sildenafil on proliferation, the epithelium from the proximal half of the colons was isolated and subjected to flow cytometry to detect the proliferative marker Ki67 (Figure 7A,B). This showed a median of 23% Ki67^+^ cells in the control animals on drinking water alone, but this was significantly reduced to 11% and 7% by sildenafil and malonyl-sildenafil treatment, respectively. Sections of the distal colon were also stained for PCNA, which showed the expected proliferative compartment in the lower third of the crypts (Figure 6C). Both sildenafil and malonyl-sildenafil reduced the number of PCNA+ cells/crypt by about 30% (Figure 6D). Taken together, these results demonstrate that orally administered malonyl-sildenafil and sildenafil were equally effective at suppressing proliferation in the colon epithelium of mice.

## 3. Discussion

Sildenafil epitomizes the hallmark Rule of 5 (Ro5) properties [29,30] regarding the overall size and lack of polarity that allows near-complete systemic delivery into the circulation [28,31]. Extensive structural studies of sildenafil have identified the piperazine nitrogen as amenable to modifications that affect pharmacokinetic properties without dramatically affecting pharmacological efficacy [32,33,34,35]. To develop a non-systemic PDE5i we chose to replace the methyl group of sildenafil with malonate, which is predicted to dramatically reduce permeability at post-gastric pH values between 5–7. This modification did not affect the inhibitor potency as malonyl-sildenafil was equally potent as sildenafil toward human PDE5a. The IC_50_ value for sildenafil determined using the assay conditions in the present study was slightly higher than previously reported [36]. However, the results are congruent with a previous study suggesting that the carboxylate mimics the phosphate of cGMP to increase affinity [33]. Our initial concern was whether sufficient levels of malonyl-sildenafil could cross the plasma membrane of cells to inhibit PDE5. To test this, we used a novel assay whereby the entry of PDE5i into colon cancer cells amplifies the GC-C activity that is driven by a low level of the synthetic agonist linaclotide. We used ELISA to measure cGMP levels in our system, which demonstrated a dramatic increase in the effective concentration (EC_50_) of malonyl-sildenafil compared to the parent compound sildenafil. This observation was supported using the activation of cGMP-dependent protein kinase (PKG2) as a readout for the ability of PDE5i to enter cells and increase cGMP levels. As the IC_50_ toward purified PDE5 was similar, the most likely interpretation is that more compound was required to break through the plasma membrane to reach the intracellular PDE5a. While these results support the idea that the carboxylic acid group impeded the diffusion of malonyl-sildenafil across the plasma membrane, this was not demonstrated empirically in the present study. Intestinal permeability leading to systemic delivery is a complex process that involves transporters and efflux pumps as well as simple diffusion across biological membranes. More detailed studies to measure apparent permeability in both apical–basolateral and basolateral–apical directions using Caco-2 monolayers will be necessary to fully understand the pharmacokinetics of malonyl-sildenafil. 

Sildenafil exhibits near-complete systemic delivery [31] and, in circulation, it undergoes hepatic modification into desmethyl-sildenafil that retains some bioactivity [37]. The prominent desmethyl-sildenafil peak in the MS/MS ion spectrum is due to the ionization energy as it was not detected in the liquid chromatography elution profile using pure compound, or in the plasma from malonyl-sildenafil treated mice (not shown). One of the most significant findings is that the oral administration of malonyl-sildenafil even at concentrations up to 36 mg/kg exhibited minimal leakage into the circulation but was excreted at high levels in the feces. This contrasted sharply with sildenafil, which peaked at 1–2 h in the plasma and was not detected in the feces. The typical human dose of sildenafil is 0.7–1.4 mg/kg, that is, equivalent to the consumption of a 50 or 100 mg pill, respectively. The 36 mg/kg dose used here is 25 times the maximum human dose and 2-fold the human dose with allometric scaling. While it is possible that our measurements of malonyl-sildenafil in plasma are underestimated due to possible first-pass metabolism, it is unlikely that pass-through would go undetected at the high doses tested here. This supports the idea that malonyl-sildenafil has low bioavailability in vivo and is largely retained in the gut lumen. The typical transit time for these mice is between 2–4 h [17], so the continued excretion of malonyl-sildenafil in the feces 8 h after oral gavage was surprising. A detailed excretion analysis was not carried out in the present study, but it was estimated that only 10–20% was recovered in the feces. It is reasonable that delayed excretion would occur if large amounts of the compound remained in the mucus and underlying mucosa. However, the generation of undetected metabolites by the intestinal epithelium is also a possibility. Regardless, this property is ostensibly ideal for a drug that targets the colon epithelium. A major drawback to the use of contemporary PDE5i for colon cancer prevention is the contraindication for patients taking nitrates to treat stable angina due to synergistic hypotensive effects [38]. It was shown here that malonyl-sildenafil did not significantly affect blood pressure in mice exposed to ISMN. This supports the pharmacokinetic data and is congruent with the notion that bioactive lipophilic metabolites were not generated from malonyl-sildenafil. The present study did not examine intestinal or hepatic metabolism of malonyl-sildenafil, but a drug–drug interaction study is an important first step in the future development of malonyl-sildenafil as an experimental drug for use in humans.

Taken together, the pharmacokinetic results demonstrated that malonyl-sildenafil did not enter the circulation, but it was unclear whether it could target the colon epithelium. Previous studies have demonstrated that the PDE5i treatment of mice can alter intestinal homeostasis by suppressing proliferation and apoptosis while promoting differentiation [17,18]. The suppression of proliferation in the preneoplastic epithelium has been proposed to mediate the CRC prevention effects of PDE5i because this proliferative compartment is where replication errors and genotoxic stress drive neoplastic transformation [39,40]. Two different approaches were used to demonstrate that oral administration of malonyl-sildenafil to mice was equally effective as sildenafil at suppressing the proliferative compartment in the colon. This result was surprising because the 18-fold reduction in cell penetration for malonyl-sildenafil (EC_50_) would predict a lesser response in vivo. It is plausible that malonyl-sildenafil levels became concentrated in the mucosal region over time. However, a more plausible explanation is that the lengthy exposure of malonyl-sildenafil to the colon epithelium, compared to the transient delivery of sildenafil by systemic delivery through the vasculature, produced a larger area under the curve to compensate. These possibilities might be explored in future studies with this compound.

There are several limitations to this study. Firstly, as discussed above, we did not directly measure the apparent permeability of malonyl-sildenafil using established Caco-2 or MDCK monolayers, and we did not carry out detailed metabolism and excretion studies. The results from such experiments could reveal unexpected alternate explanations for the observations presented. Furthermore, our results demonstrated that exposure to malonyl-sildenafil for 8 days did not have obvious deleterious effects on mice. For CRC prevention, patients would consume the compound daily over many years. While the results provide proof of principle that charged PDE5i have potential clinical utility as non-systemic drugs, they set the stage for future more elaborate pre-clinical toxicology studies that would include many months of exposure. Secondly, the suppression of proliferation was used as a pharmacodynamic marker to show that the malonyl-sildenafil could affect the gut epithelium, but conclusions regarding the ability of this compound to suppress colon cancer will warrant further investigation.

The present results illustrate the principle that existing drugs can be modified to reduce systemic bioavailability but retain pharmacological activity in the gut. The results also illustrate the first steps for screening these gut-targeted compounds, which include the demonstration that they can enter intestinal epithelial cells to affect the target while avoiding detection in the circulation. Since the goal of creating non-systemic drug analogs is to reduce toxicity and drug–drug interactions, classical absorption, distribution, metabolism, and excretion (ADME) studies are still essential next steps in drug development. The present study showed that malonyl-sildenafil did not produce any systemic metabolites PDE5 inhibitory activity, in general the intestinal metabolism of non-systemic drugs could still generate toxic metabolites, or they could affect the function of nutrient transporters and drug efflux pumps. Importantly, because the dosing of systemic drugs is derived from bioavailability and toxicity, new gut-specific dosing studies with the drug analogs will be necessary to determine pharmacodynamic plateaus to minimize systemic leakage.

In summary, with the goal of creating a non-systemic PDE5i that targets the colon epithelium, we have synthesized a novel analog of sildenafil containing a carboxylic acid group derived from malonic acid. Malonyl-sildenafil remains in the colon for many hours without entering the circulation and is equally as effective as sildenafil at suppressing epithelial proliferation. The results shown here are proof of principle for a strategy to leverage existing drugs for the prevention and treatment of intestinal disorders, with fewer side effects and drug–drug interactions.

## 4. Materials and Methods

### 4.1. Chemicals

Malonyl-sildenafil was custom-synthesized by Leadgen Labs (Orange, CT, USA) using desmethyl-sildenafil and malonic acid as building blocks. Desmethyl-sildenafil was purchased from Chemspace (Monmouth Junction, NJ, USA). The sildenafil citrate (Revatio) and Linaclotide (Linzess) were extracted from pharmacy-grade pills/capsules in water, and stored as aliquots at −80 °C. The doxycycline, puromycin, G418 and isosorbide mononitrate were from Sigma Chemical (St. Louis, MO, USA) and all other chemicals and reagents were from Fisher Scientific (Waltham, MA, USA).

### 4.2. Western Blotting

Cells were lysed by incubation in ice-cold lysis buffer (50 mM Tris-HCl pH 7.4, 150 mM NaCl, 1% Nonidet P-40, 0.25% deoxycholate) supplemented with protease and phosphatase inhibitor cocktail (Fisher Scientific, Waltham, MA, USA). Clarified protein extracts were quantified using a BCA protein quantification kit (Fisher Scientific, Waltham, MA, USA). Proteins were then separated on 10% polyacrylamide minigels (BioRad, Hercules CA, USA), transferred to nitrocellulose membranes and blocked with 5% BSA in PBS containing 0.025% Tween 20. Antibodies used for immunoblotting were β-actin (1:2000, Sigma) and PKG2 (mAb E7; 1:200, Santa Cruz Biotechnology). Antibodies against phospho-VASP (Ser239) and total VASP were from Cell Signaling (1:1000; Danvers, MA, USA). Blots were visualized on film using Pierce-ECL substrate.

### 4.3. In Vitro Pharmacology and Permeability Assays

The predicted LogS and LogD at different pH values for each structure were calculated using tools available at Chemicalize.com. The PDE-5 inhibitory activity of sildenafil and malonyl-sildenafil was measured in vitro using a commercially available kit according to the manufacturer instructions (BPS Bioscience, San Diego, CA, USA). The assay was reproduced in three independent experiments with triplicate wells in each replicate. IC_50_ values were calculated using curve fitting functions in GraphPad Prism 9 (San Diego, CA, USA).

The cell-based permeability assays were performed using LS-174T colon cancer cells obtained from the American Type Culture Collection. Cells were treated with 1 µM linaclotide together with different PDE5i concentrations for 4 h, followed by processing of cell extracts for cellular cGMP measurement using cGMP ELISA kits according to the manufacturer instructions (Cayman Chemical, Ann Arbor, MI, USA). The calculated EC_50_ values are the concentration of compound required to increase cellular cGMP levels by 50% of maximum based on predictions determined by curve fitting in GraphPad Prism 9 (San Diego, CA, USA). To assess activation of cGMP signaling by immunoblotting, we used LS174T cells with inducible PKG2 expression that were described recently [41]. 

### 4.4. Animal Studies 

The goal of the studies described herein is to develop first-in-class agents for the chemoprevention of colorectal cancer in humans. Rodents are widely regarded as essential models for early pharmacology and toxicology studies, and mice were chosen owing to extensive published data demonstrating CRC chemoprevention by conventional PDE5i. For all animal studies, six-week-old C57BL6 mice were purchased from Jackson Laboratories (Bar Harbor, ME), and allowed to acclimate to the Augusta University animal facility for at least 2 weeks prior to experimentation. Mice were housed in shoebox cages with 5 mice of the same sex per cage with water and irradiated rodent chow (Teklad 8904) ad libitum. Cages were maintained in a conventional specific pathogen-free environment at 24 °C, and with a 12 h light/dark cycle. On completion of experiments or at any signs of morbidity, mice were euthanized by controlled CO_2_ delivery for humane asphyxiation followed by decapitation. The experimental design included equal numbers of randomly chosen male and female mice for the proof-of-principle pharmacokinetic studies, but only male mice were used for the surgical procedure and for the colon proliferation studies since our previous studies have found no difference in the response to cGMP elevation on colon homeostasis or colon cancer prevention. Statistical analysis was only undertaken for studies where each group size was at least n = 5. All studies carried out with animals were approved by the Augusta University Institutional Animal Care and Use Committee. Descriptions of animal studies are compliant with the ARRIVE guidelines [42] and recommendations made by the British Journal of Pharmacology [43]. 

### 4.5. Pharmacokinetic Studies

Mice were randomly caged in groups of 3 of each sex where each data point (dose or time) was derived from 6 mice. Stocks of sildenafil and malonyl-sildenafil were prepared in diH20 to deliver different doses by gavage of 0.1 mL per mouse, or 9 mg/kg for the time course. Blood was drawn from mice by submandibular bleed, and plasma was isolated using BD microtainer lithium tubes (Becton, Dickinson and Company, Franklin Lakes, NJ, USA). Fecal pellets were collected over a 2 h period and the PDE5i were extracted by mixing with 1 mL methanol for 3 s in a Pro-200 homogenizer (Cole-Parmer, Vernon Hills, IL, USA), followed by centrifugation at 10,000× *g* for 10 min.

### 4.6. Arterial Blood Pressure Measurements

Male C57BL/6J mice (n = 4) were implanted with a PA-C10 blood pressure transmitter in the right carotid artery (Data Sciences International, St. Paul, MN, USA), and after recovery from surgery, the data was recorded using the DSI PhysioTel system (Data Sciences International, St. Paul, MN, USA). For each experiment, the mice were randomly selected for treatment with ISMN mixed in diH20, and 0.1 mL was injected i.p. for a final dose of 300 mg/kg. This was followed by gavage of 0.1 mL of H_2_0 (control), sildenafil or malonyl-sildenafil at 10 mg/kg, and the mice were returned to the cage for monitoring. The experiment was repeated until each treatment had been reproduced in 5 independent biological trials after which mice were euthanized.

The surgical procedures and implantable telemetry approach for measuring blood pressure in mice have been described in detail previously [44]. In brief, all surgical instruments were autoclaved on a stainless-steel tray that was used as the sterile surface during surgery. The mouse is anesthetized in an isoflurane-inducing chamber. Anesthesia used a fitted nose cone connected to an isoflurane system running 2% in a stream of 100% oxygen mixed with room air. The incision area was shaved and prepped with betadine and rinsed with 70% ethanol. Anesthesia was assessed continuously during shaving and prepping for surgery, with appropriate adjustments. An approximately 10 mm incision was made in the ventral neck region over the trachea to expose the left carotid artery. A subcutaneous tunnel was made on the right side leading to the mid scapular region where a sterile Data Sciences BP transmitter unit was inserted into a pocket, and the catheter tip from the unit was inserted into the carotid artery and secured in place with silk ties. The incision was closed in one layer with sterile, 5-0 or 6-0 silk suture. After closing the incision, the mouse was removed from anesthesia, buprenorphine SR was administered i.p. and the mouse was placed in a clean, warmed cage to recover. Once fully conscious and able to ambulate, mice were transferred to individual plastic shoe box cages and placed on top of Data Sciences receivers that have been matched to the respective transmitters. Approximately 5–7 days were allowed from surgical procedures before beginning experimental procedures and measurements. 

### 4.7. Measuring Cell Proliferation in the Colon Epithelium

Mice (n = 6, males) were provided sildenafil or malonyl-sildenafil ad libitum in the drinking water dissolved at a concentration that delivered 9 mg/kg daily. Mice were weighed daily, and after at least 5 days of continuous consumption they were sacrificed, and the colon was dissected and processed for quantitation of proliferation by flow cytometry (proximal region) and by IHC (distal region). For flow cytometry, the epithelium was enriched by isolating crypts as described previously [45], followed by ethanol fixation and staining with phycoerythrin-conjugated anti-mouse Ki-67 antibodies according to the manufacturer instructions (Biolegend, San Diego, CA, USA). Flow cytometry was carried out using a BD Accuri C6 flow cytometer, and data were analyzed using FCS Express 7 software (De Novo Software, Pasadena, CA, USA). Processing the distal colon tissues for histology and immunohistochemistry was carried out as previously described. Briefly, tissues were fixed with 10% formaldehyde, embedded in paraffin blocks, and sectioned by the Augusta University histology core. The tissues were probed using antibodies to PCNA (1:100; Cell Signaling, Danvers, MA, USA). Visualization of PCNA antibodies was carried out using the ImmunoCruz ABC kit (Santa Cruz Biotechnology, Dallas, TX, USA). At least 10 different regions of the colon containing approximately 8 crypts per region were counted for each mouse. For both the flow-cytometer and immunohistochemistry approaches, each group (control, sildenafil, malonyl-sildenafil) contained 5 mice (male) and the experiment was reproduced in 2 independent replicates.

### 4.8. LC-MS/MS Analysis

Samples were prepared by adding 180 µL acetonitrile (containing 1 nM loperamide as internal standard) to 20 µL mouse plasma or fecal extract. Samples were vortexed for 5 min at room temperature and centrifuged at 16,000× *g* for 10 min at 4 °C. After transfer of the supernatant to a glass vial, 8µL was injected into a Thermo Hypersil C18 column (100 × 2.1 mm, 1.9 µm) on a Shimadzu Nexera UHPLC system, and the separation of compounds was performed using gradient elution from 25% to 95% acetonitrile (with 0.1% formic acid) in 5 min at a flow rate of 0.2 mL/min and oven temperature of 50 °C. The effluent was ionized using positive ion electrospray on a Thermo TSQ Quantiva triple-quadrupole mass spectrometer. The integrated peak areas for these transitions were calculated for each sample using Skyline software (version 21.0, University of Washington). The concentration of compounds in the samples was determined using a linear equation derived from running a set of 7 standards. The limit of detection was approximately 1 nM for each compound.

### 4.9. Statistical Analysis

All data are expressed as mean ± SD, and individual data points are shown in all studies where statistical significance was tested (n ≥ 5). The group sizes (number of mice per group or independent experimental replicates) for each study are stated in the figure legends. Cell proliferation and blood pressure data were analyzed for significance using one-way ANOVA followed by Tukey’s multiple comparisons test comparing to the untreated control. The differences were considered to be significant when *p* < 0.05. All statistical analyses were carried out using GraphPad Prism 9 (San Diego, CA, USA).

## Figures and Tables

**Figure 1 ijms-24-09397-f001:**
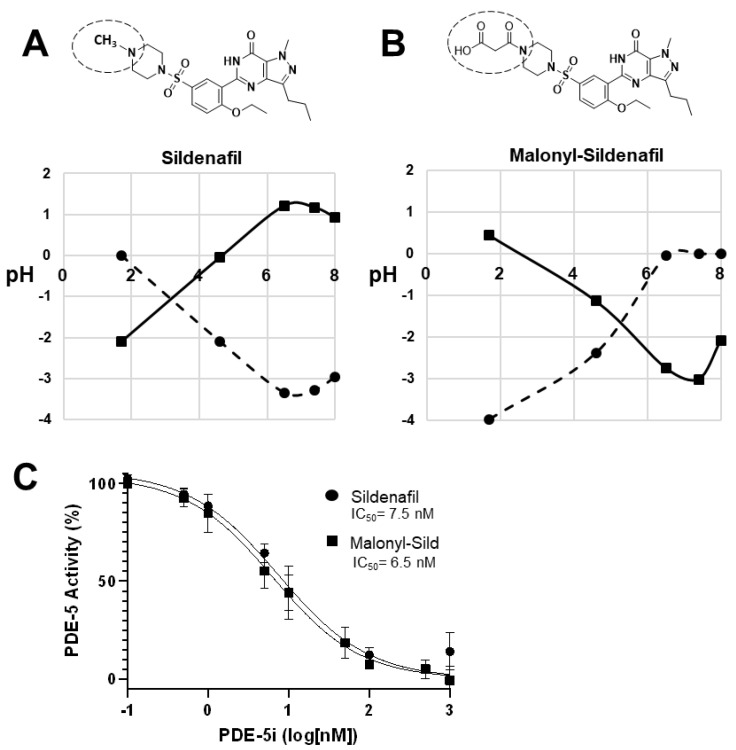
Malonyl-sildenafil is a polar analog with predicted low permeability. (**A**) Sildenafil is largely hydrophobic and exhibits poor solubility (LogS, circles) and high permeability (LogD, squares) at physiological pH. (**B**) Substitution of the piperazine methyl group in sildenafil with malonic acid is predicted to increase solubility and reduce permeability at physiological pH. The substitution is indicated by a dashed circle. LogS values from −2 to −4 are poorly soluble, and LogD values 0 to 5 are permeable. (**C**) In vitro assay for PDE5 inhibitory activity of sildenafil and malonyl-sildenafil. Data shown are mean +/− SD (*n* = 3).

**Figure 2 ijms-24-09397-f002:**
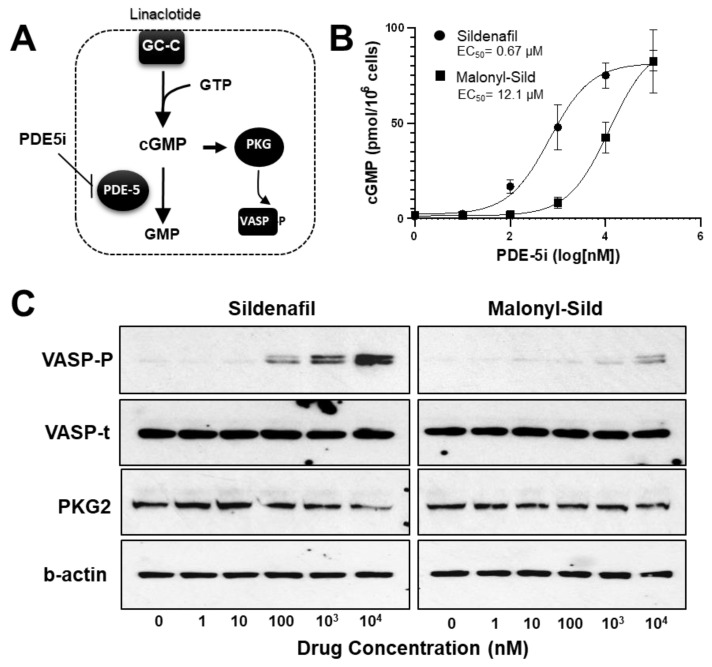
Malonyl-sildenafil retains pharmacological activity with less potency in colon epithelial cells indicating poor membrane permeability. (**A**) Cartoon depicting an LS-174T-ll cell-based assay where movement of the PDE5 inhibitor (PDE5i) across the plasma-membrane (dashed line) leads to increased cGMP in colon cancer cells treated with a low dose of linaclotide to stimulate guanylate cyclase-C (GC-C). Increased cGMP levels activate cGMP-dependent protein kinase (PKG) that phosphorylates vasodilator stimulated phosphoprotein (VASP). (**B**) The effective concentration (EC_50_) of sildenafil and malonyl-sildenafil was determined based on the ability to enter LA-174T cells to increase cGMP levels. (**C**) The LS174T-ll system (as depicted above) was treated with increasing doses of sildenafil or malonyl-sildenafil, and VASP phosphorylated on Ser239 (VASP-P) was determined by immunoblotting. Total VASP (VASP-t) and β-actin are loading controls, whereas PKG2 indicates equal expression of ectopic PKG2. Data in panel A are mean +/− SD (*n* = 4), and results in panel C are representative of 3 independent experiments.

**Figure 3 ijms-24-09397-f003:**
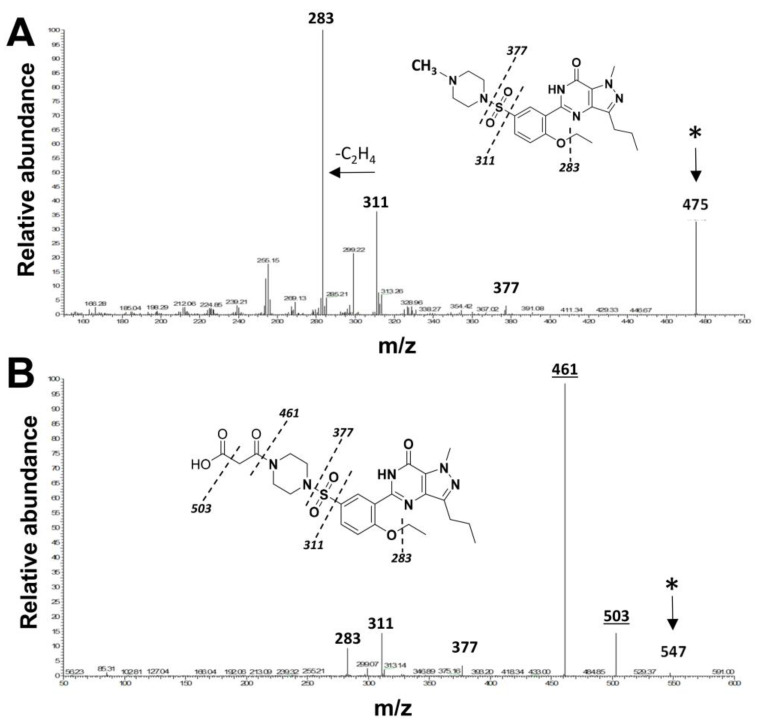
MS/MS Spectra for sildenafil and malonyl-sildenafil. Ion fragment patterns for sildenafil (**A**) and malonyl-sildenafil (**B**). Proposed peak identification is illustrated by the dashed lines on the inset structure diagrams. The enlarged bold text indicates common ion peaks between the two compounds, and the asterisk marks the parent compound. The underlined peaks in panel (**B**) indicate unique ions generated by the malonyl group.

**Figure 4 ijms-24-09397-f004:**
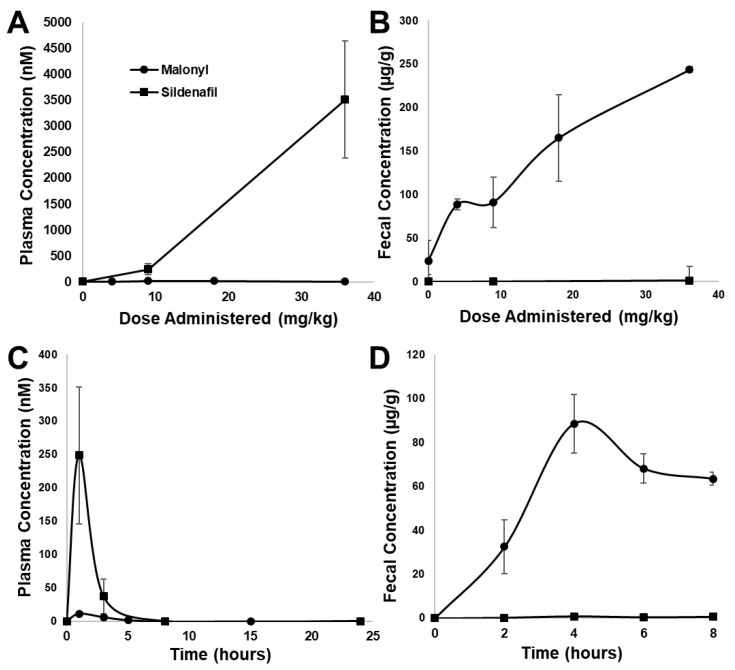
Pharmacokinetic analysis of malonyl-sildenafil in mice. Increasing doses of malonyl-sildenafil (circles) or sildenafil (squares) were administered to mice by gavage. LC-MS/MS was used to measure the compounds in plasma after 1 h (**A**), or in feces collected between 2–4 h (**B**). The compounds were also measured at different time points after administration of a single 9 mg/kg dose in plasma (**C**) and feces (**D**). Data are mean +/− SD (*n* = 6).

**Figure 5 ijms-24-09397-f005:**
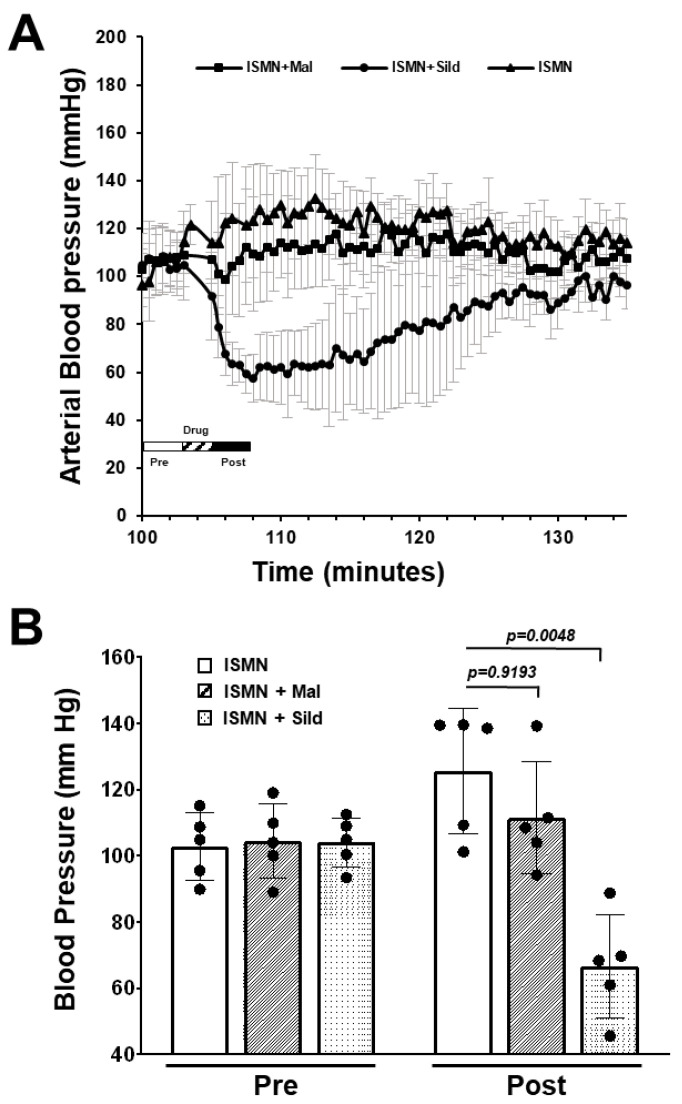
Malonyl-sildenafil does not interact with isosorbide mononitrate in mice. Mice were implanted with a carotid artery blood pressure transmitter (Data Science, PA-C10) and real time telemetry was recorded with a DSI PhysioTel system. Mice were randomly administered isosorbide mononitrate by i.p. injection (ISMN; 300 mg/kg) with orally administered vehicle, sildenafil or malonyl-sildenafil (10 mg/kg) as indicated. (**A**) Blood pressure recording with inset key showing when the drugs were administered. (**B**) Mean pressure (3 min) before drug (pre) and after drug (post). Data are mean +/− SD (*n* = 5), and *p* values are from ANOVA with Tukey’s multiple comparisons test.

**Figure 6 ijms-24-09397-f006:**
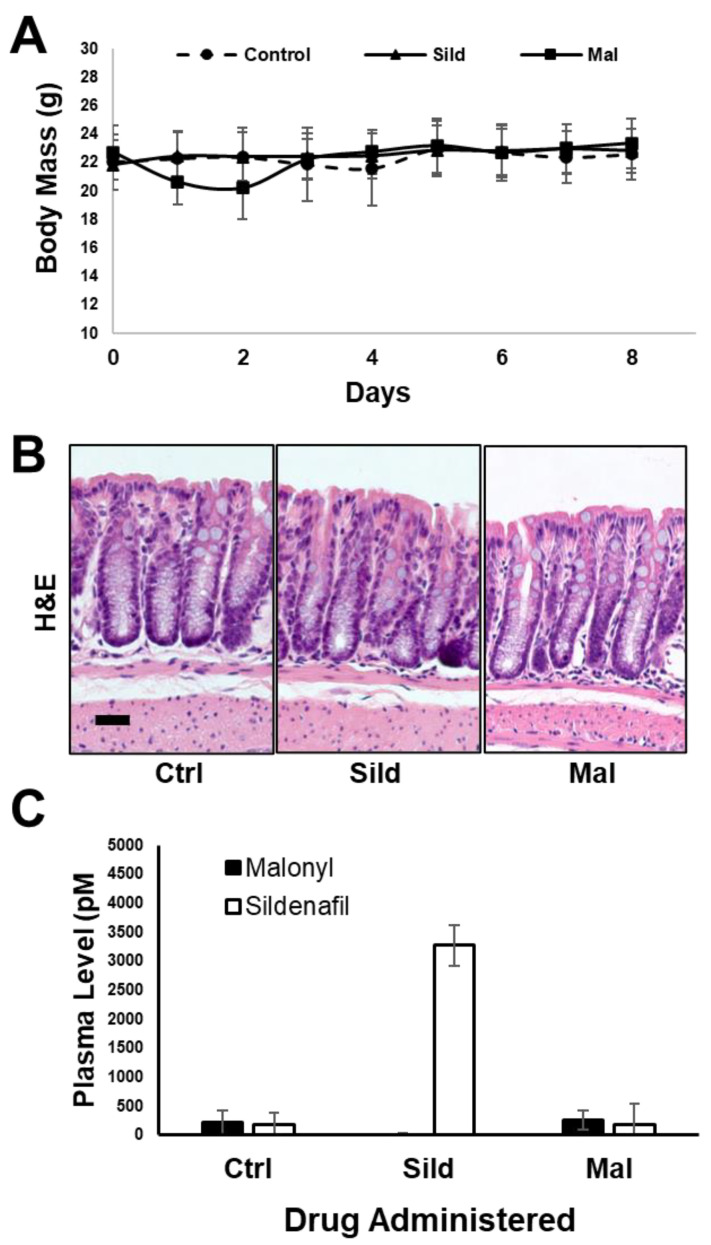
Malonyl-sildenafil is well tolerated in mice. Mice were maintained on water alone (Ctrl), sildenafil (Sild) or malonyl-sildenafil (Mal) dissolved in the drinking water for up to 8 days as indicated. (**A**) Body mass measurements, (**B**) representative H&E-stained sections of the distal colon, scale bar is 30 μ. (**C**) Detection of Sild or Mal in the plasma at 8 days. Data are mean +/− SD (*n* = 5).

**Figure 7 ijms-24-09397-f007:**
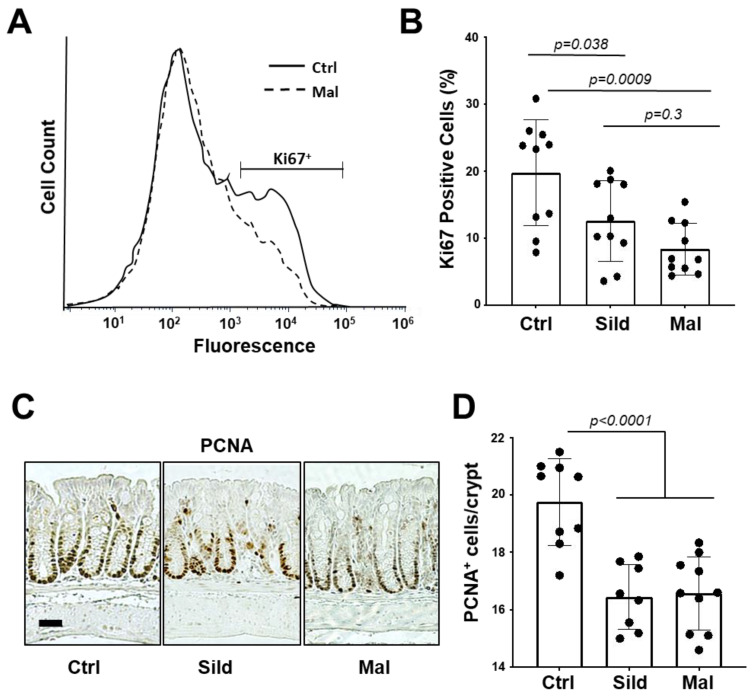
Malonyl-sildenafil suppresses proliferation in the colon epithelium of mice. Mice were treated with 9 mg/kg sildenafil (Sild) or malonyl-sildenafil (Mal) in the drinking water, or with water alone (Ctrl). (**A**) After 8 days the proximal colon epithelium was harvested and subjected to flow cytometry measurement of Ki67^+^ cells. (**B**) Quantitation of treatment effects on Ki67^+^ cells. (**C**) PCNA+ cells in the distal colon were visualized by IHC, the scale bar is 30 μ. (**D**) Quantitation of treatment effects on PCNA+ cells per crypt. The plots in panels B and D show mean +/− SD, and *p* values are from one-way ANOVA followed by Tukey’s multiple comparisons test.

## Data Availability

Data are contained within the article.

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
