# Peer review of "A Non-Systemic Phosphodiesterase-5 Inhibitor Suppresses Colon Proliferation in Mice"

_ijms, 2023, doi:10.3390/ijms24119397_

Round 1
Reviewer 1 Report
Lee et al. conducted a study titled “A non-systemic phosphodiesterase-5 inhibitor suppresses colon proliferation in mice.”. Overall, the paper describes the development of an analog of the phosphodiesterase-5 inhibitor (PDE5i) sildenafil, which has reduced lipophilicity and potential for fewer side-effects and drug interactions, and is being investigated for repurposing as a colon cancer chemopreventive agent. The analog, malonyl-sildenafil, was found to have similar pharmacology as sildenafil, showed negligible levels in mouse plasma after oral administration but was detected at high levels in the feces. The authors demonstrate the effectiveness of malonyl-sildenafil in suppressing colon epithelium proliferation in mice, which is consistent with the effects of conventional PDE5i. The information presented in the present article is interesting and novel.
I have the following comments:
- As the authors state, sildenafil has potential to suppress tumor initiation, but does not affect the size of initiated polyps. In clinical practice, the drug would most likely used as a long-term prophylactic/preventative drug in patients at high risk of developing CRC. Hence, safety data over prolonged use is of critical importance for translation into clinical practice. Was ‘long-term’ toxicity/safety data collected for malonyl-sildenafil? The authors may also include this in the discussion section of the manuscript.
- Figure 6 displays detection of Sild or Mal in plasma at 8 days. It appears as if both are present in the Ctrl group. Mal is more abundant in Ctrl than in Sild group. Please clarify.
- Please clarify determination of ‘EC50’ in the methods section
- Please include limitations in the discussion section.
Author Response
Reviewer 1.
Comment 1. Was “long-term” toxicity/safety data collected for malonyl-sildenafil? Could be included in the discussion.
Response: Thank you for your insightful commentary on our study. You are correct in that our goal is to provide a preventative agent for individuals at high risk for CRC development, and it is expected that they would need to take the compound daily for years to maintain the inhibitory effect. Due to ample data demonstrating the long-term safety of Tadalafil taken daily by men to treat benign prostate hyperplasia, we are optimistic that a non-systemic analog will be even more benign. In the present study we only superficially tested safety (e.g., body mass and gross histology in mice) after taking for a week, but we are carrying out longer-term studies in CRC prevention models where the mice will take malonyl-sildenafil daily for several months (equivalent to 5-6 human years). We are cognizant of the fact that regulatory agencies such as the FDA and EMA would require more extensive toxicology studies before approval. We have added additional text in the discussion section to elaborate on this issue.
Comment 2. Figure 6 displays detection of Sild or Mal in plasma at 8 days…
Response: The objective for this experiment was to determine whether malonyl-sildenafil might accumulate in the plasma over time. The results showed barely detectable malonyl-sildenafil, and very small amounts of sildenafil (3.5 nM). It is noteworthy that even the sildenafil would not register on the scale shown in Figure 4. Note that the scale in Figure 6 is in pM and we attributed the low levels of malonyl-sildenafil to be essentially noise rather than signal. However, the reviewer’s query is reasonable, so we re-examined the data to determine how the malonyl-sildenafil levels were calculated. We determined that the levels reported in the specimens from the control animals were slightly higher due to background determinations of adjacent samples. Between each set of specimens, we ran blanks (solvent only), and the background levels in the blanks were typically negligible but for the dataset under consideration, the levels of compound were so low that the levels in the blank became more significant. We re-analyzed this dataset by subtracting an average of the blanks for each group and replaced this panel in Figure 6. The results are very similar and do not affect the conclusions, we appreciate the reviewer for bringing this to our attention as it is a more accurate representation that has improved the quality of our results.
Comment 3. Please clarify determination of “EC50” in the methods section.
Response: This has been done in the revised manuscript.
Comment 4. Please include limitations in the discussion section.
Response: This has been done in the revised manuscript.
Reviewer 2 Report
The authors decribed in this manuscript their attempt to design a non-systemic PDE5 inhibitor for the prevention of colon cancer and showed promising data.
Minor issues:
- Line 184 and 185: The text seems to refer to the wrong figure (obvious should be 7C and 7D instead of 6C and 6D)
- Figure 6C: Why are there measurable levels of Sildenafil and Malonylsildenafil in control animals and measurable levels of Sildenafil in animals treated with Malonylsildenafil?
Major issues
- Based on its structure one could assume that Malonylsildenafil could be metabolized to N-Desmethylsildenafil. Instead of measuring the potential metabolite in PK samples, the authors chose to test drug combinations in a telemetry study. This was problematic from two perspectives: It would be much more informative to measure the PK samples and it would make to additional animal study obsolete. In addition, a number of in vitro experiments (e.g. in vitro metabolite identification experiment with hepatocytes or liver microsomes or intestinal microsomes, cellular permeability assay with Caco-2 cells) coud be performed to inform whether a PK study is necessary
- Although high concentrations of Malonylsildenafil were detected in feces of treated mice, the total recovery of the drug in feces was not determined. Based on values shown in Fig. 4B and 4D, the total recovery could be less than 10% of total drug amount dosed to the animal. This would raise the question what happened to the drug. The authors speculated that drug could stick to intestinal mucosa. Another explanation could be a fast first-pass extraction by the liver and subsequent biliary excretion of the drug.
- Although some efficacy could be demonstrated for Malonylsildenafil with drinking water formulation, this was of less relevance for human situation, because the primary dosing route would be very like oral. Additionally, an oral dosing in the animal study could verify authors' hypothesis that the drug sticks for very long time to the intestinal mucosa.
Author Response
Reviewer 2.
Minor issues:
Comment 1: reference to the wrong figure in results (7C, D instead of ^C and 6D)
Response: Thank you for bringing this to our attention, it has been fixed in the revised manuscript.
Comment 2: Why was there measurable levels of the PDE5i in the control animals?
Response: This issue has been addressed in detail in response to reviewer 1, but the short answer is that the levels shown are mostly “noise” due to the scale used (pM compared to nM in Figure 4). However, we have re-analyzed the data to make it slightly more precise and have replaced panel C in Figure 6.
Major issues:
Comment 1: Based on the structure one could assume that malonyl-sildenafil could be metabolized to desmethyl-sildenafil…
Response: We are grateful to this reviewer for the insightful comments. We completely agree that malonyl-sildenafil looks as though it could be metabolized to generate desmethyl-sildenafil but this did not occur in vivo following oral administration. As stated in the 2nd paragraph of the discussion, the energy used for fragmentation during the LC-MS/MS resulted in a peak at 461 m/z corresponding to desmethyl-sildenafil but it is important to recall that desmethyl-sildenafil is a unique structure with a different LC-elution time during the LC-MS, and should not be confused with the 461 m/z peak in the malonyl-sildenafil fragmentation signature. Our first experiments looked for desmethyl-sildenafil in the plasma but in contrast to mice treated with sildenafil it was never detected. This is likely due to the fact that the 1st pass enzymes more readily detect the lipophilic compounds such as the piperazine methyl group on sildenafil and are less likely to recognize the carboxyl group in malonyl-sildenafil.
This reviewer also suggested that in vitro metabolite identification studies are warranted, and we fully agree. In fact, we are currently funded to carry out those studies to further develop this compound, but the results of those experiments would be unclear owing to a lack of information about permeability and pharmacological activity of any metabolites detected. We argue that our innovative blood-pressure study was more informative because it screened for any potential metabolite that exhibited both PDE5i inhibitory activity as well as permeability. This is a critical experiment for the current study because our premise is that reduced systemic delivery would reduce the likelihood of drug-drug interactions.
Comment 2: Although high concentration of Malonyl-sildenafil were detected in feces…
Response: We agree that a complete accounting of the administered compound will need to be completed for ultimate approval of this compound for clinical testing, but that is far beyond the purpose of this manuscript. Again, we are truly grateful for this reviewers expertise in drug development by raising this issue. We were also interested in determining if there was near-complete elimination in the feces, but it was not possible because we stopped collecting at 8 hours post gavage. This is more than double the transit time for these animals, but malonyl-sildenafil was still being shed at high concentrations. We would need to collect feces for at least a 24 h period, but even then it is complicated by shedding of luminal epithelium during that time. The suggestion by the reviewer that the fecal malonyl-sildenafil might be subject to first pass metabolism by the liver and subsequent biliary excretion is highly unlikely because (1) we would have detected malonyl-sildenafil in the plasma, (2) any hepatic metabolism would result in a compound undetected by our LC-MS/MS signature.
Comment 3: Malonyl-sildenafil drinking water formulation…relevance for human situation…
Response: Oral administration of malonyl-sildenafil is an essential component of our thesis, in that the compound stays luminally and does not enter the circulation. Most of the studies shown here were performed by oral gavage because timing is essential for both PK studies. The longer PD study provided the drug orally by allowing the animals to drink the compound in the water, which was more precise dosing (we measured volume consumed) compared to providing in the food. Both approaches to oral administration are completely relevant to the ultimate human application.
Reviewer 3 Report
Please read the attachment. Thank you.

Author Response
Reviewer 3.
Comment 1: Suggest more information about the significance of CRC as a health burden be included in the introduction.
Response: The 1st ten lines of the introduction highlight in a concise manner the clinical need for chemoprevention in CRC, the target population (e.g., high-risk patients), and limitations of the current standard of care. We did not elaborate further because this manuscript is not directly testing colon cancer inhibition, but is more focused on drug development.
Comment 2: The methods section could be improved by providing more information about the animal models used and the ethical considerations involved in the study.
Response: We respectfully submit that we have detailed the need for animal use and have provided great detail regarding their treatment and manipulation for each model. It is unclear what else this reviewer is looking for. As stated in the Methods section, our experimental design and reporting of experiments using animals is consistent with the internationally respected ARRIVE guidelines, and the ethical use has been approved by our institutional committee (as stated in the methods section).
Comment 3: The authors could further discuss the potential clinical implications of their findings, such as the possible use of malonyl-sildenafil as a first-in-class drug for colon cancer prevention.
Response: We have added some discussion describing some limitations of the present study that elaborate on the next steps for drug development.
Other questions/comments: (response is italicized)
⎯ Title: Please delete the dot at the end of the title.
A period was already included in the original submission.
⎯ Keywords: Please provide between 5 and 10 keywords that should not repeat the words/phrases that appeared in the manuscript title.
Keywords present in the title are acceptable, but we have added an additional word.
⎯ At the end of the introduction section, please add a paragraph that introduces the outline of the manuscript.
We expanded the outline sentence in the original submission to make a short paragraph as requested.
⎯ Section 4 Materials and methods should appear after Section 1.
⎯ Section 2. Results should move to Section 4.
We used the IJMS template to format our submission according to the journal guidelines, and this journal prefers the Methods section to be after the discussion (section 4).
⎯ All figures should be mentioned or explained in the text.
This was done in the original submission, except for an error in which discussion of Figure 7 referred to Fig. 6, but this was corrected.
⎯ What are the current limitations of conventional PDE5i for colon cancer prevention?
See the second-to-last paragraph of the introduction.
⎯ How does the modification of sildenafil with malonic acid reduce its lipophilicity, and what is the potential impact of this modification on drug efficacy?
See Figure 1.
⎯ How does malonyl-sildenafil affect cellular cGMP levels compared to
conventional sildenafil?
See Figure 2.
⎯ What is the pharmacokinetic profile of malonyl-sildenafil in mice, and how does it differ from conventional sildenafil?
See Figure 3.
⎯ What is the impact of malonyl-sildenafil on colon epithelium proliferation, and how does this compare to the effects of conventional PDE5i?
See Figure 7.
⎯ How does modifying sildenafil with malonic acid provide a novel approach to developing a first-in-class drug for colon cancer chemoprevention?
There is currently nothing for CRC chemoprevention, so our compound could be developed into a first-in-class drug for this indication.
⎯ What are the potential clinical implications of using malonyl-sildenafil for colon cancer prevention, and how could this approach be further developed in future research?
Future studies will test the chemopreventive ability in murine CRC models (currently being carried out), and the compound will need required ADME and toxicology studies to move it forward as an experimental drug. Except for some text highlighting limitations of the current study (e.g., that we used colon proliferation as a surrogate for possible CRC prevention), we did not add additional explanatory text to the manuscript.
Round 2
Reviewer 2 Report
Minor issue:
Figure 6C is still misleading. The authors should indicate in legend the limits of quantification for Sildenafil and Malonyl Sildenafil, and, if these are "noises", indicate also that the exposure values of in control animals are belong quantification. Same for the groups treated with drugs.
Major issues:
The authors' responses were not convincing for the following reasons:
1. The authors used a cellular potency shift assay to assess the permeability of Malonyl sildenafil. Though potency shift in a cellular assay is suggestive for lower permeability, the correlation between potency shift (in this case 18-fold between Sildenafil and Malonyl sildenafil) and permeability is not quantitative. It remains unclear how low is the permeability of Malonyl sildenafil. The fact that the compound did show clear concentration-dependent effect on an intracellular target indicates rather an acceptable permeability. Since the intestinal mucosa has a very large surface area, even compounds with very low permeability (e.g. Atenolol) can have a fraction absorbed as high as 50-60% in human. The in vitro results shown here are not sufficient to demonstrate that Malonyl Sildenafil is not penetrating gut wall.
2. The low systemic exposure of Malonyl sildenafil could be explained in different ways: Chemical instability, metabolic degradation in intestinal lumen or in enterocytes, high first-pass extraction by the liver and subsequent excretion into bile, etc. All these reasons could lead to diminisched pharmacological effects of the drug, both systemically and topically. Without knowing the fate of the administered amount of the drug, low systemic exposure is of low relevance for the purpose of proof of principle.
3. Since the pharmacological active metabolite N-desmethyl sildenafil was not detected in plasma after the oral administration of Malonyl sildenafil and the exposure of Malonyl sildenafil was very low, the telemetry study didn't provide any additional information and was dispensable. In light of reducing unnecessary animal experiments (3R) this could have been avoided.
4. The recovery of the administered drug amount in feces is too low, even when extrapolated to 24 h, to explain the systemic non-availability of the compound. The extended excretion in feces could well be explained by strong first-pass hepatic extraction and a subsequent slow biliary excretion of the drug in unchanged form. A very low systemic exposure due to relative slow intestinal absorption and fast first-pass hepatic extraction is not unusual after oral administration.
5. The clinical relevance of the PD experiment with drinking water fromulation (which is very unlikely for human) is limited because the pharmacokinetic profile of the drug (systemically or topically) would be completely different compared to oral administration (which is clinically more relevant), this would very likely lead to different PD efficacy. If the authors are convinced that Malonyl sildenafil stays in intestinal mucosa for very long time without being absorbed after an oral administration, there is no reason to use a drinking water formulation. Furthermore, a comparison of drinking water formulation and oral administration could be even very informative regarding the fate of the administered drug amount.
Author Response
Minor issue:
Figure 6C is still misleading. The authors should indicate in legend the limits of quantification for Sildenafil and Malonyl Sildenafil, and, if these are "noises", indicate also that the exposure values of in control animals are belong quantification. Same for the groups treated with drugs.
Response: We agree with the reviewer that the panel is misleading because we should not detect sildenafil or malonyl-sildenafil in the control animals that have never been exposed to it. The data are that way because a linear equation derived from a standard curve was used to calculate the levels of the compounds in plasma. The line went through the origin despite the lower concentrations reading at background, which reflected a limit of detection around 1 nM. The equation still produced low numbers despite the actual readings being at baseline. We have added more text in the in the Materials and Methods section that clarifies the panel.
Major issues:
The authors' responses were not convincing for the following reasons:
1. The authors used a cellular potency shift assay to assess the permeability of Malonyl sildenafil. Though potency shift in a cellular assay is suggestive for lower permeability, the correlation between potency shift (in this case 18-fold between Sildenafil and Malonyl sildenafil) and permeability is not quantitative. It remains unclear how low is the permeability of Malonyl sildenafil. The fact that the compound did show clear concentration-dependent effect on an intracellular target indicates rather an acceptable permeability. Since the intestinal mucosa has a very large surface area, even compounds with very low permeability (e.g. Atenolol) can have a fraction absorbed as high as 50-60% in human. The in vitro results shown here are not sufficient to demonstrate that Malonyl Sildenafil is not penetrating gut wall.
Response: We genuinely appreciate the expertise and time spent by this reviewer to consider our work. We completely agree with the statement that our cell permeability studies are not equivalent to Caco-2 apparent permeability (Papp) assays; they measure different things. We have included additional text in the revised discussion that addresses this point. It is important to note that our goal was not to demonstrate a reduced Papp because the addition of a carboxylic acid group to the parent compound would certainly reduce permeability, and the overall structural changes are predicted to confer poor bioavailability compared to sildenafil (see SwissADME reports below). Rather, our main concern was whether the anionic analog was still able to get into cells at levels sufficient to affect intracellular cGMP. The reviewers comment regarding “acceptable permeability” underscores the utility of our model because there is a fine line between being able to enter the epithelium and getting into the circulation. We acknowledge the concept they present using Atenolol as an example, which is why we went directly to the animal models for the preliminary PK studies shown rather than carry out extensive ADME studies. We do not believe that Papp measurements are necessary to draw the conclusions that we have from our results. The second issue raised here is that the in vitro results shown are insufficient to show that our compound is not penetrating the gut wall. We fully agree, which is why we carried out the PK studies in mice.
2. The low systemic exposure of Malonyl sildenafil could be explained in different ways: Chemical instability, metabolic degradation in intestinal lumen or in enterocytes, high first-pass extraction by the liver and subsequent excretion into bile, etc. All these reasons could lead to diminisched pharmacological effects of the drug, both systemically and topically. Without knowing the fate of the administered amount of the drug, low systemic exposure is of low relevance for the purpose of proof of principle.
Response: We agree with the reviewer that the scenarios are possible, and have acknowledged the ideas in the revised discussion along with arguments highlighting why they are much less likely based on the results presented.
3. Since the pharmacological active metabolite N-desmethyl sildenafil was not detected in plasma after the oral administration of Malonyl sildenafil and the exposure of Malonyl sildenafil was very low, the telemetry study didn't provide any additional information and was dispensable. In light of reducing unnecessary animal experiments (3R) this could have been avoided.
Response: We respectfully disagree with the reviewer on this point. As the reviewer stated, desmethyl sildenafil was not detected in the malonyl-sildenafil treated animals. However, it remained possible that metabolism of our compound by the intestinal epithelium could result in a metabolite other than desmethyl-sildenafil (that we were not looking for) that could retain pharmacological activity and enter the circulation. This is very unlikely because the effect of metabolism is typically to reduce lipophilicity, but we carried out these experiments because they were essential to support the conclusions that we made. If an unknown biologically active metabolite derived from malonyl-sildenafil was able to enter the circulation undetected, then it might have been responsible for the effect on epithelial proliferation that we observed. This experiment was essential because it demonstrated a lack of such metabolites.
4. The recovery of the administered drug amount in feces is too low, even when extrapolated to 24 h, to explain the systemic non-availability of the compound. The extended excretion in feces could well be explained by strong first-pass hepatic extraction and a subsequent slow biliary excretion of the drug in unchanged form. A very low systemic exposure due to relative slow intestinal absorption and fast first-pass hepatic extraction is not unusual after oral administration.
Response: We agree with the reviewer that this scenario is possible, and have acknowledged it in the revised discussion along with arguments highlighting why it is much less likely based on the results presented.
5. The clinical relevance of the PD experiment with drinking water fromulation (which is very unlikely for human) is limited because the pharmacokinetic profile of the drug (systemically or topically) would be completely different compared to oral administration (which is clinically more relevant), this would very likely lead to different PD efficacy. If the authors are convinced that Malonyl sildenafil stays in intestinal mucosa for very long time without being absorbed after an oral administration, there is no reason to use a drinking water formulation. Furthermore, a comparison of drinking water formulation and oral administration could be even very informative regarding the fate of the administered drug amount.
Response: We regret that we don’t understand the point being made here, as we equate consumption of the compound dissolved in the drinking water (no specific formulation) as an oral administration. The PK studies were done using gavage, but the longer-term PD study was done by including the compound in the drinking water of the mice. To our understanding, these are both clinically relevant modes of “oral administration” in rodents. We could have used gavage on the PD study, but we felt that this would expose the animals to unnecessary stress.
Sildenafil

Malonyl-sildenafil

Atenolol

Round 3
Reviewer 2 Report
The revised manuscript and authors' responses didn't address the issues raised by the reviewer satisfyingly. The pivotal statement "We show that this compound exhibits poor cell-permeability,..." was not supported by the available results shown here. As stated in the previous review, potency shift assay in a cellular model is not an appropriate surrogate for cellular permeability measurement. Lack of systemic exposure of a compound administrated orally is not a proof for lack of penetration of intestinal epithelia. Very low total recovery in feces is in clear contradiction to the hypothesis that the compound is not absorbed. Without knowing the fate of the compound in GI tract it is impossible to judge the suitability of the compound as a potential drug. Finally, the observed effects in vivo are of less relevance due to the formulation with drinking water (different PK profiles after oral gavage and via drinking water formulation will have different pharmacological effects).
Although the results described in this manuscript are of high novelty and could potentially be of high interest for further investigation, the relevance of the results are strongly limited due to missing data, which was not acknowledged sufficiently in the discussion.
Author Response
It is unfortunate that we are unable to satisfy this reviewer's concerns. The point of this study is not so much to show the discovery of a new drug to treat disease as it is to highlight the concept that existing drugs designed for systemic delivery can be modified by altering lipophilicity to target the gut mucosa, which is more efficient than engineering a new drug starting with the target. We demonstrate this for a single compound, that like all potential drug candidates will need extensive ADME studies to determine where the drug goes, how it is metabolized, and how it is excreted as required before any Federal drug safety agency permits first in human studies. However, this is a manuscript and not an investigational new drug application. All the data presented support the hypothesis, but as the reviewer indicates, it is not 100%. It is very important to acknowledge that great science generates additional questions to inspire further investigation, and that is what our study does. We understand the additional questions that are sticking points with the reviewer. Firstly, they feel that our assay for pharmacological activity of the compound that is dependent upon entry into the intestinal epithelial cells is not the same as diffusion through Caco-2 or MDCK cells. However, the principles of cell-entry vs barrier are similar in that they both are dependent upon diffusion across the plasma membrane. There is surely no argument that replacing a methyl-group with a negatively charged group will impede diffusion across the lipid bilayer. Because we are interested in the possibility of further developing malonyl-sildenafil as a drug candidate, we are currently engaged in carrying out Papp studies but this will take many months. Even with experimentally-derived Papp values (A-B and B-A), it won't significantly move the needle for the present study because permeability is dependent upon whether it is a substrate for the efflux transporters, and in this case also monocarboxylate transporters. Therefore Federal drug safety agencies with require many additional studies with independent transporter/pump expression, inhibitors, effect on transport of known substrates, all of which will need to be done to fully understand PK of our compound. While the cell line we used is not widely used for drug development, they are more suitable for the present study because unlike intestinal epithelium, Caco-2 cells do not express guanylate cyclase C or PKG2. Therefore our approach allowed us to measure pharmacology with crossing the plasma membrane as the limiting factor (that includes transporters and pumps). The second issue raised by the reviewer is that we did not determine in the present study the metabolism and excretion routes for our compound and therefore can not with 100% certainty state that is was not absorbed systemically. As we have stated, the fact that we detect approximately 20%/24hr of the administered compound in the feces and do not detect it in the plasma strongly supports the idea that it had poor systemic bioavailability. The most likely scenario to account for the rest of the compound administered is that there is a reservoir in the epithelium and colon mucosa that will be slowly shed into the lumen over several days as part of epithelial turnover, but it is also possible that intestinal metabolism could generate metabolites that are similarly shed in the stool. Again, Federal drug safety agencies will require extensive metabolism and biodistribution studies, but while these will add to the safety profile, they are unlikely to significantly affect our interpretation of the results presented in our manuscript.